# Molecular and Genetic Characterization of Hepatitis B Virus (HBV) among Saudi Chronically HBV-Infected Individuals

**DOI:** 10.3390/v15020458

**Published:** 2023-02-06

**Authors:** Mariantonietta Di Stefano, Giuseppina Faleo, Thomas Leitner, Wei Zheng, Yang Zhang, Amna Hassan, Marwan J. Alwazzeh, Josè R. Fiore, Mona Ismail, Teresa A. Santantonio

**Affiliations:** 1Department of Clinical and Surgical Sciences, Section of Infectious Diseases, University of Foggia, 71122 Foggia, Italy; 2Theoretical Biology & Biophysics Group, Los Alamos National Laboratory, Los Alamos, NM 87544, USA; 3Department of Computational Medicine and Bioinformatics, University of Michigan, Ann Arbor, MI 48109, USA; 4Department of Biological Chemistry, University of Michigan, Ann Arbor, MI 48109, USA; 5Division of Gastroenterology, King Fahd Hospital of the University, Al-Khobar 31952, Saudi Arabia; 6College of Medicine, Imam Abdulrahman Bin Faisal University, Dammam 34221, Saudi Arabia; 7Infectious Disease Division, King Fahd Hospital of the University, Al-Khobar 34217, Saudi Arabia

**Keywords:** hepatitis B virus (HBV), genotype D, HBV resistance, major hydrophilic region (MHR), HBsAg

## Abstract

The study aimed to characterize the genotype and subgenotypes of HBV circulating in Saudi Arabia, the presence of clinically relevant mutations possibly associated with resistance to antivirals or immune escape phenomena, and the possible impact of mutations in the structural characteristics of HBV polymerase. Plasma samples from 12 Saudi Arabian HBV-infected patients were analyzed using an in-house PCR method and direct sequencing. Saudi patients were infected with mainly subgenotype D1. A number of mutations in the RT gene (correlated to antiviral resistance) and within and outside the major hydrophilic region of the S gene (claimed to influence immunogenicity and be related to immune escape) were observed in almost all patients. Furthermore, the presence of mutations in the S region caused a change in the tertiary structure of the protein compared with the consensus region. Clinical manifestations of HBV infection may change dramatically as a result of viral and host factors: the study of mutations and protein-associated cofactors might define possible aspects relevant for the natural and therapeutic history of HBV infection.

## 1. Introduction

Human hepatitis B virus (HBV) is an enveloped DNA virus belonging to the Hepadnaviridae family [1] associated with a wide spectrum of clinical manifestations resulting in both acute and chronic liver infection [2]; in addition, chronic HBV hepatitis can be complicated by liver cirrhosis, and hepatocellular carcinoma [3,4]. Despite the wide implementation of immunization programs, HBV infection still remains a major public health problem with approximately 296 million people who are chronically infected worldwide [5,6,7].

The incidence rate of hepatitis B in Saudi Arabia has declined from 19.65 in 2003 to 13.63 per 100,000 inhabitants in 2016 [8]. Furthermore, a recent cross-sectional study involving 74,662 participants reported a prevalence rate of hepatitis B infection of approximately 1.3% [8]. Nevertheless, hepatitis B continues to be a serious health problem in Saudi Arabia [9], mainly affecting older people with advanced forms of liver disease and several co-morbidities [10]. HBV, despite being a DNA virus, is characterized by a high genetic heterogeneity due to the peculiar mechanism of viral replication requiring the activity of reverse transcriptase (RT) polymerase for the reverse transcription of an RNA intermediate, called pregenomic RNA (pgRNA) [11].

Consequently, there are 10 genotypes of HBV(A–J) until now and several subgenotypes [12]. The predominant genotype in the Eastern Mediterranean countries including Saudi Arabia is genotype D.

Additionally, HBV has a quasispecies distribution in infected individuals; mutations may occur in all of the genes and can be responsible for treatment resistance, immune escape, disease outcome, and carcinogenesis [13,14,15,16].

Mutations in the reverse transcriptase (RT) region of HBV polymerase can reduce the susceptibility to antiviral drugs or the restoration of replicative fitness [17]. In addition, HBV surface antigen (HBsAg) contains the major hydrophilic region (MHR), a dominant epitope crucial for binding to neutralizing antibodies. The presence of mutations in this region might change the hydrophilicity, electric charge, or acidity of the loop, having a number of pathobiological effects on the structure of HBV polymerase. Moreover, due to overlapping between the RT gene and HBsAg, the presence of mutations can also contribute to reduce the binding affinity for neutralizing antibodies, including those induced by HBV vaccine [11].

This study aimed to assess the HBV genotypes and subgenotypes circulating in Saudi Arabia and to investigate the presence of clinically relevant mutations in RT and S region in order to verify the possible impact of mutations on the structural characteristics of HBV polymerase in Saudi infected patients.

## 2. Patients, Material, and Methods

### 2.1. Study Population

In this study, we included hepatitis-B-infected patients older than 18 years attending the Hepatology Clinic in the King Fahad Hospital of the University-Al-Khobar (Saudi Arabia) enrolled in the period January to March 2018. The patients were included in the study, after signed informed consent, if they were reactive for HBsAg for more than six months and if they had detectable plasma HBV DNA, while we excluded patients with undetectable plasma HBV DNA, co-infection with HCV or HDV, autoimmune liver disease, primary biliary cirrhosis, hemochromatosis, alpha one antitrypsin deficiency, Wilson’s disease, liver cirrhosis, or hepatic injury caused by drug use; pregnant women and nursing mothers were excluded as well.

The clinical and demographical data were obtained from the patient’s medical records and included: age, sex, and history of previous use of antiviral treatment. The standard clinical investigations included liver function tests (ALT, alanine aminotransferase; AST, aspartate aminotransferase; ALP, alkaline phosphatase; total bilirubin; albumin); platelet count; international normalized ratio (INR); and results of HBV serological markers (HBsAg, HBeAg, anti-HBs, anti-HBc, and anti-HBe) using the Abbott Architect assay (Abbott Diagnostics Division Max-Planck-ring 2, 65205, Wiesbaden, Germany).

Transient elastography (TE) (FibroScan^®^ 502 Touch, Echosens, Paris, France) was used to assess liver fibrosis (liver stiffness measurement), expressed in kpascal (kPa) in combination with controlled attenuation parameter (CAP), used to determine liver steatosis, expressed in decibels per meter (dB/m).

All TE procedures were performed by an experienced practitioner (M.I.) after at least 4 h of fasting by the patients. The measurements were made on the right lobe of the liver, as described by the manufacturing company, and ten successful measurements were obtained for each patient. TE failure was recorded when no value was obtained after at least ten attempts. The results were considered unreliable if the number of valid attempts was fewer than 10, the success rate was <60%, or the interquartile range/median was >30%.

### 2.2. Plasma Samples

The plasma samples were stored at −80 °C until testing. Plasma samples were obtained from 26 Saudi Arabian HBV-infected patients.

The real-time HBV PCR viral load was performed using the Artus^®^ HBV RG PCR kit (QIAGEN GmbH, QIAGEN Strasse 1, 40724 Hilden, Germany). The lower detection limit was 10 IU/mL.

### 2.3. HBV Sequencing and Genotyping

HBV genotype and detection of mutations in the polymerase gene were carried out using an in-house PCR method, as shown elsewhere [18]. For each patient, HBV reverse transcriptase (HBV-RT) (344 amino acids) and the overlapped HBsAg (226 amino acids) full-length sequencing was performed on plasma samples, as described previously [18,19].

Briefly, HBV DNA was extracted from 140 uL of plasma using a commercially available kit (QIAmp DNA blood mini kit, Qiagen Inc., 19300 Germantown Rd., Germantown, MD 20874, USA); the first round of PCR was carried out in the final volume of 50 uL containing Ampli Taq Gold polymerase enzyme and using the following primer pairs: 5-GGTCACCATATTCTTGGGAA-3′ and 5′-GTGGGGGTTGCGTCAGCAAA-3′. PCR conditions were: one cycle at 93 °C for 12 min, 40 cycles (94 °C 50 s, 53 °C 50 s, 72 °C 1 min and 30 s), and a final cycle at 72 °C for 10 min. [18]. When the first amplification round provided negative results, a second round of PCR was used. The second round of PCR was carried out using 5 μL of the first PCR product under the same condition as the first round of PCR. The PCR products were electrophoresed on 2% agarose gel and stained with ethidium bromide [18].

The lower detection limit of the described nested PCR in this study was estimated to be 20 copies/mL HBV DNA.

PCR products were sequenced by using eight different overlapping sequence-specific primers with a Big Dye terminator v. 3.1 cycle sequencing kit (Applied Biosystems, Foster City, CA, USA) with the Sanger sequencing method, and an automated sequencer (ABI-3100), as reported elsewhere [18,19]. The sequences were analyzed using Seqscape-v.2.0 software. Amino acid (aa) polymorphisms associated with drug resistance were obtained using geno2pheno HBV. Genotyping was carried out by phylogenetic comparison of all patient sequences (RT and S genomic regions separately) with genotype reference sequences, as recommended by Schaefer [20], that used one reference sequence per subgenotype, labelled in the trees as “GT-X.accession number”, plus a non-human primate virus as out group; 7 + 1 sequences. In addition, BLASTN identified the closest previously published sequences in GenBank that were added to phylogeny; sometimes, the same GenBank sequence was identified as closest to >1 of our sequences. Thus, this resulted in addition 11 and 7 GenBank sequences to the S and RT trees. We did use the SH-test in PhyML to assess reconstruction robustness. All GenBank sequences were subgenotyped as genotype D1 in agreement with their original classification, providing adequate subgenotyping robustness. Sequences were aligned using MAFFT V7 under the G-INS-1 algorithm [21], and phylogenetic trees were calculated using PhyML V3 under a GTR + I + G model and best of NNI + SPR search [22] (Figure 1 and Figure 2).

The online ExPASy ProtParam tool [23], available at http://expasy.org/tools/protparam.html, (accessed on 2 September 2022) was used to study, in both the S gene and the RT gene, molecular weight, theoretical isoelectric point (pI), extinction coefficient, aliphatic index, instability index, grand average of hydropathy (GRAVY), and the total number of positive and negative residual amino acids [23].

The online I-TASSER (I-TASSER Suite 5.2Department of Computational Medicine and Bioinformatics, Department of Biological Chemistry, University of Michigan Medical School, 100 Washtenaw Avenue, Ann Arbor, MI 48109-2218, USA) server was used for automated protein-structure prediction and structure-based function annotation [24,25,26].

## 3. Results

Demographic, virological, and clinical characteristics of the twenty-six HBV-infected Saudi Arabian patients enrolled in the study are shown in Table 1. Patients were predominantly males (17 were male and 9 were female), with a mean age of 45.2 years (range 31–57). All patients were anti-HBe positive with HBV DNA levels ranging from 1 × 10^3^ to 6 × 10^6^ IU/mL. Liver fibrosis was generally mild (F1–F2), and only three patients were treated with antiretrovirals (tenofovir disoproxil fumarate) (Table 1).

Although, at the time of enrollment in the study, all tested patients had detectable HBV-DNA in plasma, for molecular investigation, only plasma samples obtained from 12 of the patients were suitable for HBV DNA extraction and amplification. For the remaining samples, the absence of HBV DNA was confirmed even after running the second round of PCR amplification.

The phylogenetic tree analysis of the HBV polymerase gene showed that all of the 12 subjects were infected with an HBV D genotype. In particular, we observed that eleven isolates belonged to subgenotype D1 (92%), and one to subgenotype D2 (8%). No deletion or insertion were detected in the polymerase region. However, several amino acid (aa) substitutions were observed in the RT region of HBV polymerase and in the S gene (Table 2 and Table 3).

In the RT gene, the Y135S substitution was observed in 11/12 HBV strains, followed by the N248H substitution found in 10/12 HBV isolates regardless of the HBV subgenotype.

The change of serine to threonine at position 213 was identified in two HBV isolates (Table 2). Only in one HBV isolate (patient 772), we identified a mutation at position 181 (A181G), known to confer resistance to adefovir and tenofovir as well as telbivudine and lamivudine [27,28,29].

Additionally, this strain also showed a mutation at position 233 (I233M) known to be associated with resistance to adefovir, although this resistance included a valine as aa change [30]. The mutation at position 215 associated with resistance to adefovir was observed in 5/12 HBV strains, although two HBV strains showed a modified glutamine in serine, two in histidine, and one in proline. Interestingly, all patients were naïve to the treatment. Other substitutions were also seen at positions 54, 122, 266, 329, 336, and 337 (Table 2).

The presence of aa substitutions at different positions in the surface region was observed in all HBV isolates. At position 207, different aa changes were found in 6/12 HBV strains; the substitutions T118A and L209V were observed in two different HBV isolates (Table 3) and the isolate (HBV_800 strain) showed several aa mutations at positions 129, 131, 133, 193, 203, 204, 205, and 209, including those associated with hepatitis B surface antigen escape. Our study showed mutations within the major hydrophilic region (MHR) of HBsAg in the second loop of “a” determinant associated with escape from vaccine-induced immunity in several HBV isolates (Table 3). A mutation at position 129 (Q129H) was observed in three isolates and additional mutations T131N + M133T were found in the HBV_800 strain, whereas the HBV_272 strain showed more additional mutations at position 109, 120, 126, and 131. The HBV_738 isolate displayed a mutation at position 144 (D144E).

The presence of polymorphisms in the region of the S protein outside the MHR (aa 78-aa 99) associated with antigenicity prediction was observed in 4/12 (33%) of HBV Saudi strains (001; 800; 170; 389).

The physicochemical properties of both polymerase and S genes of HBV isolates are reported in Table 4 and Table 5, as observed by the online ExPASy ProtParam tool (SIB Swiss Institute of Bioinformatics, Quartier Sorge—Batiment Amphipole, 1015 Lausanne, Switzerland [23]. The theoretical isoelectric point (pI) in the polymerase protein was found to be either acidic or neutral in all samples ranging from 5.11 to 7.39 (Table 4).

The GRAVY score was indeed similar for all polymerase proteins except one; in fact, 11/12 had proteins with a GRAVY score above 0, whereas, in an HBV isolate, the GRAVY score was below 0 (−0.034) which is considered a hydrophilic protein (globular protein). Our sequences of the polymerase gene with an instability index of more than 44.55 resulted in unstable proteins.

For the S gene, our sequences were identified as alkaline proteins in 11/12 HBV isolates with a pI value above 8.2. However, for the HBV_272 isolate, the pI was not computed because the sequence had multiple polymorphisms. The GRAVY index was greater than 0 in all S proteins, showing a more probable membraneous protein (hydrophobic protein). The instability indices around 51.98–64.12 identified all S proteins as unstable (Table 5).

Tertiary structures of HBsAg from HBV consensus and from HBV_800 (an isolate that showed several mutations) were predicted by I-Tasser and further by DeepFold models (Figure 3, Figure 4 and Figure 5).

Our results based on DeepFold models showed that the two proteins seemed to have completely different folds with TM-score = 0.38. The two models both have N-terminus helix region, C-terminus helix region, and a centrally beta-like core region; however, the orientations between those two helix regions of the two models are quite different, mainly due to the mutation on core region changing the fold orientation of the N terminus region and C terminus region (Figure 3).

## 4. Discussion

Although HBV is a DNA virus, it is characterized by a great variability, with different genotypes and subgenotypes and also an intraindividual variability, probably influencing several aspects of the infection, including prognosis, evolution, response/resistance to antivirals, and sensitivity to natural or vaccine/induced neutralizing antibodies

In this study, we characterized the genotypes and subgenotypes of HBV circulating in Saudi Arabia, the presence of clinically relevant mutations possibly associated with resistance to antivirals or immune escape phenomena, and the possible impact of mutations in the structural characteristics of HBV polymerase.

Phylogenetic analysis showed that all Saudi subjects were infected with HBV genotype D in line with other molecular studies in Saudi Arabia [31,32,33,34] This finding corresponds to the epidemiological profile observed in other neighboring countries [35,36]. Interestingly, in fact, the D1 subgenotype is the dominant subtype in the Mediterranean area [32], whereas the presence of D2 subgenotypes had been found in different geographic areas as Europe, India, Australia, and Africa [37,38,39,40].

In our study, 11 of the HBV isolates studied belonged to the subgenotype D1 (92%) and one to the subgenotype D2 (8%). These results support the subgenotype profile already presented in the recent study using a limited number of isolates from Saudi Arabia [34].

In this paper, different mutations in the RT region of the polymerase peptide as well as in the surface region were observed in all of the HBV Saudi strains.

Two different mutations, Y135S and N248H, in the reverse transcriptase region of the polymerase peptide have been identified among our HBV strains causing infection in the Saudi population. Tuteja and collaborators showed both these two mutations in the HBV strains with the D genotype circulating in the infected Indian population, and the frequency was 67 vs. 59, respectively [29]. We found a mutation at position 248 in RT in 83% of Saudi HBV isolates, independently of D subgenotypes. This is consistent with a study by Chavan and collaborators, in which this mutation was described as the most common genotypic variant in their West Indian population [41].

The substitution in the RT region of serine with threonine at position 213 (S213T) was observed and detected in naïve patients with subgenotype D1. The S213T in the RT region of the polymerase gene was reported to be associated with HBV A2, B, and C genotypes in previous studies [42,43]; moreover, Zhang and colleagues [43] described the S213T mutation and classified it as an unconventional mutation since it is found in patients with virological breakthrough and treated with ADV, ETV, and LMV.

The 772_HBV isolate, from an infected individual naïve to antiviral treatments, exhibited an aa mutation at position 181 (A181G), known to confer resistance to adefovir, tenofovir, telbivudine, and lamivudine [29,44]. Another mutation also associated with adefovir resistance was observed at position 233 (I233M), though this resistance includes valine as an aa modification.

Five out of twelve HBV isolates showed the presence of a mutation at position 215 (Q215S) associated with adefovir resistance, although all these patients were naïve to the treatment.

Sequence analysis of the S gene showed aa substitutions in various positions, including those associated with hepatitis B surface antigen escape [45,46] in several HBV isolates.

Mutations in “a” determinant of surface protein were further observed in 4/12 of HBV isolates of Saudi patients; in previous reports [45,46,47], the presence of mutations in this region correlated with the absence of HBsAg in samples, though in our case we could not confirm this finding.

However, if different mutations or combinations could be involved in the absence /or presence of HBsAg is still unclear.

In our study, the presence of polymorphisms in the region of S protein outside the MHR (78 aa–99 aa) was observed in 4/12 (33%) of Saudi HBV strains (001; 800; 170; 389). It is known that this region might influence the immunogenicity and antigenicity of HBsAg, as reported by Khodadad [47].

The physicochemical properties of HBs protein confirmed the data published by Khodadad [47], showing that all the S proteins were basic, having a pI higher than 8; furthermore, the instability index around 51.98–64.12 identified the S proteins as unstable, and they were recognized as hydrophobic having a GRAVY above 0.

It is known that the clinical manifestations of HBV infection may change radically due to either viral or host factors; therefore, it may be of great importance to study, understand, and define these potentially important co-factors.

Surely, one of the limits of this study was that not all of our HBV samples could be amplified and sequenced. One possible explanation could be that sample storage could have affected DNA extraction/amplification, because of viral DNA degradation.

In conclusion, we characterized in this study the HBV strains circulating in Saudi Arabia, both from the molecular and physicochemical point of view, with the aim of providing new information needed to better clarify the potential outcomes relevant to the natural and therapeutic history of HBV infection.

## Figures and Tables

**Figure 1 viruses-15-00458-f001:**
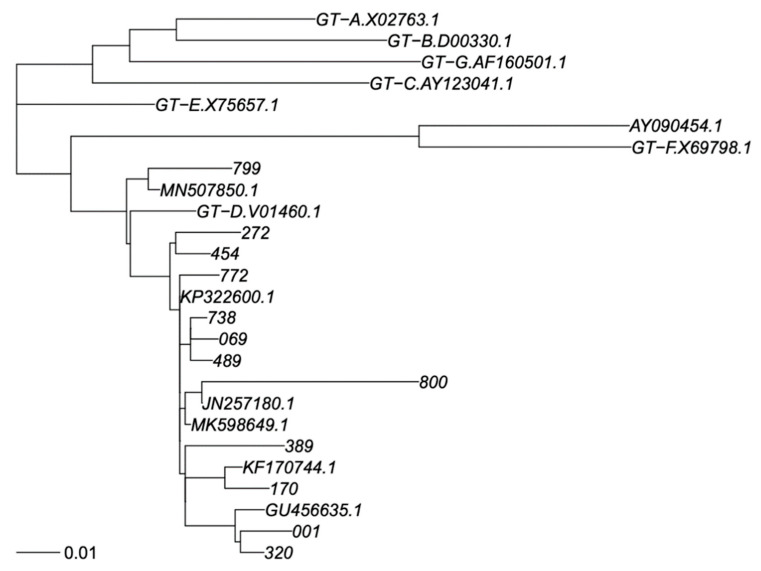
Phylogenetic tree of RT sequence fragment. Legend: Phylogenetic tree for genotype classification of RT segment. Genotyping reference sequences are labelled “GT-X”, where X indicates genotype, followed by the GenBank accession number. Our new sequences are labelled with a three-digit numerical code, and the closest sequences in GenBank are labelled with their accession number. The scalebar is in units of substitutions/site. Trees were constructed as described in Materials and Methods.

**Figure 2 viruses-15-00458-f002:**
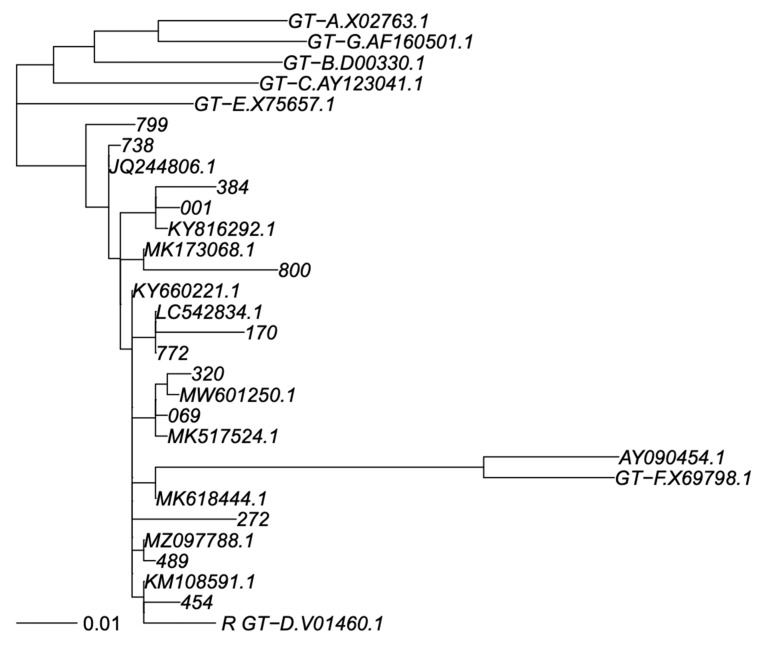
Phylogenetic tree of S sequence fragment. Legend: Phylogenetic tree for genotype classification of S segment. Genotyping reference sequences are labelled “GT-X”, where X indicates genotype, followed by the GenBank accession number. Our new sequences are labelled with a three-digit numerical code, and the closest sequences in GenBank are labelled with their accession number. The scalebar is in units of substitutions/site. Trees were constructed as described in Materials and Methods.

**Figure 3 viruses-15-00458-f003:**
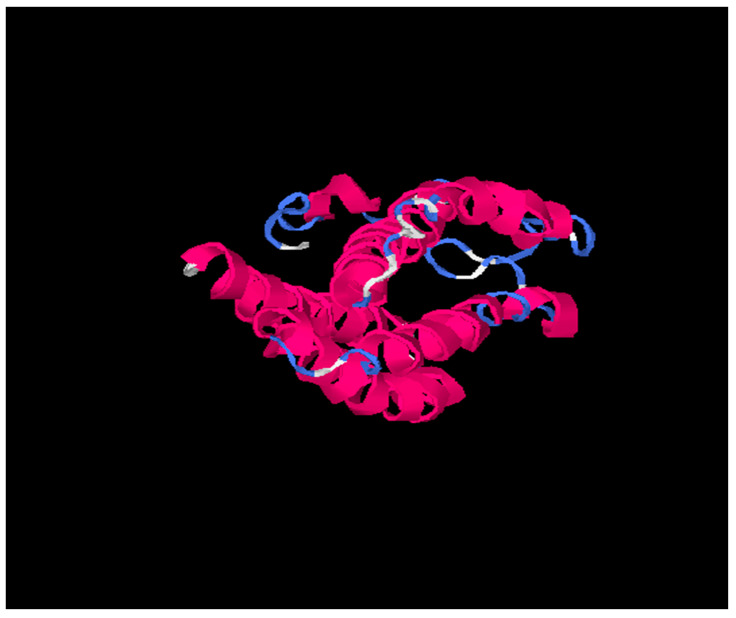
703029 (HBV Consensus) tertiary structure using I-Tasser online software. Pink is alpha-helix; Blue is coil and white region is extended strand.

**Figure 4 viruses-15-00458-f004:**
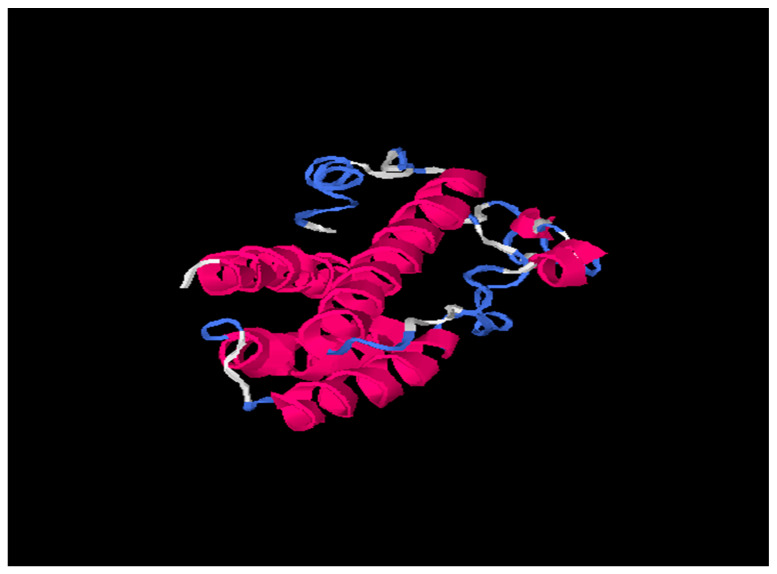
701347 (HBV_800 isolate) tertiary structure using I-Tasser online software. Pink is alpha-helix; Blue is coil and white region is extended strand.

**Figure 5 viruses-15-00458-f005:**
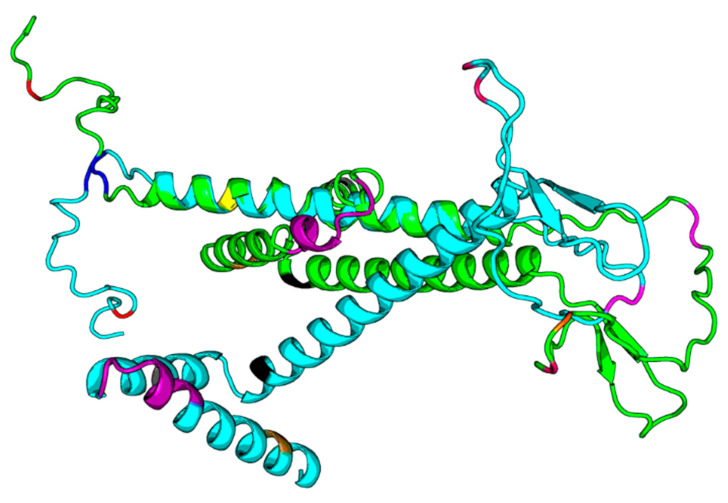
DF292. DF293.png. Legend: The green structure in the png file is DF292 (HBV consensus), and the cyan one is DF293 (HBV_800 isolate), and we highlight the sequence differences in the structure model with different colors. The two models both have N-terminus helix regions and C-terminus helix regions; however, the orientations between those two helix regions of the two models are quite different.

**Table 1 viruses-15-00458-t001:** Demographic, virological, and clinical characteristics of patients infected with HBV.

Patients	Age	Gender	Nationality	Therapy	HBsAg	HBeAg	anti-HBe	HBV-DNA	Fibrosis	Steatosis
								UI/mL	Stage	Stage
800	56	M	Saudi	No	Pos	Neg	Pos	1000	F1	S3
772	54	M	Saudi	No	Pos	Neg	Pos	1350	F0	S1
738	42	M	Saudi	No	Pos	Neg	Pos	1750	F0	S2
489	32	F	Saudi	No	Pos	Neg	Pos	ND	F1	S0
454	47	M	Saudi	No	Pos	Neg	Pos	4300	F0	S2
384	34	M	Saudi	No	Pos	Neg	Pos	6050	F2	S3
320	48	F	Saudi	No	Pos	Neg	Pos	3100	F0	S0
170	52	M	Saudi	No	Pos	Neg	Pos	1750	F1	S2
069	38	M	Saudi	No	Pos	Nd	Nd	6,000,000	Nd	Nd
001	54	F	Saudi	No	Pos	Neg	Pos	1000	F0	S2
272	39	M	Saudi	Tenofovir	Pos	Neg	Pos	4,000,000	Nd	Nd
799	31	F	Saudi	No	Pos	Neg	Pos	1000	F1	S0
883	32	F	Saudi	No	Pos	Neg	Pos	25,000	F1	S0
278	35	M	Saudi	No	Pos	Neg	Pos	13,000	F1	S3
082	39	F	Saudi	No	Pos	Neg	Pos	2000	F1	S0
156	55	F	Saudi	Tenofovir	Pos	Neg	Pos	1700	F1	S1
671	57	M	Saudi	No	Pos	Neg	Pos	1600	F1	S0
956	48	M	Saudi	No	Pos	Neg	Pos	1950	F1	S3
739	41	F	Saudi	Tenofovir	Pos	Neg	Pos	1100	F2	S0
067	56	M	Saudi	No	Pos	Neg	Pos	1050	F0	S1
561	39	M	Saudi	No	Pos	Neg	Pos	1665	F1	S1
118	38	M	Saudi	No	Pos	Neg	Pos	1245	F1	S0
470	50	M	Saudi	No	Pos	Neg	Pos	1000	F2	S1
866	54	F	Saudi	No	Pos	Neg	Pos	Nd	Nd	Nd
305	49	M	Saudi	No	Pos	Neg	Pos	1067	F1	S1
980	56	M	Saudi	No	Pos	Neg	Pos	1780	Nd	Nd

**Table 2 viruses-15-00458-t002:** Distribution of polymorphisms found for RT of HBV polymerase gene.

Sample	Genotype	Polymorphisms											
		54	122	135	181	213	215	233	248	266	329	336	337
800	D1			Y135S								L336M	
799	D1			Y135S									
772	D1			Y135S	A181G			I233M	N248H	I266L	A329T		
738	D1			Y135S					N248H				
489	D1			Y135S					N248H				
454	D1	Y54T		Y135S			Q215P		N248H				
384	D1			Y135S		S213T	Q215H		N248H				N337H
320	D1	Y54H		Y135S			Q215S		N248H	I266L			
069	D1			Y135S			Q215S		N248H	I266R			
001	D1	Y54T	F122V	Y135S		S213ST	Q215H		N248H	I266L		L336M	
170	D2		F122V						N248H	I266V			
272	D1		F122FLPS	Y135S					N248H	I266R			

**Table 3 viruses-15-00458-t003:** Distribution of polymorphisms found for the surface gene of HBV.

Sample	Genotype	Polymorphisms														
		109	118	120	126	129	131	133	144	193	203	204	205	206	207	209

800	D1					Q129H	T131N	M133T		S193L	P203L	S204N	L205P			L209V
799	D1					Q129H										
772	D1													Y206C		
738	D1								D144E							
489	D1															
454	D1						T131K			S193LS		S204T		Y206C	S207R	
384	D1											S204R		Y206S	S207T	
320	D1											S204N			S207R	
069	D1													Y206H	S207R	
001	D1											S204R		Y206C	S207T	
170	D2		T118A												S207N	
272	D1	L109R		P120A	T126IS	Q129HR	T131N									

**Table 4 viruses-15-00458-t004:** Molecular weight, theoretical isoelectric point (pI), extinction coefficient, aliphatic index, instability index, grand average of hydropathy (GRAVY), and total number of positive and negative residues of RT of the polymerase region.

	Molecular Weight	pI *	Extinction Coefficient	Aliphatic Index	Instability Index	GRAVY	+ Residual AA	− Residual AA
800	78,974	7.20	31,125	21.70	54.51	0.712	4	0
799	62,364	5.11	12,740	21.11	53.40	0.691	0	0
772	77,832	7.39	17,480	20.79	52.21	0.695	6	0
738	76,138	5.06	19,875	21.21	54.95	0.731	0	0
489	77,892	5.05	15,000	21.67	53.63	0.754	0	0
454	81,319	6.80	30,470	21.01	53.05	0.710	2	0
384	81,446	6.55	15,750	20.77	56.00	0.736	1	0
320	81,622	6.57	15,375	20.51	52.66	0.708	1	0
170	75,168	6.83	20,865	20.76	51.93	0.793	2	0
069	78,789	6.47	26,335	21.13	53.99	0.729	1	0
001	85,798	7.34	26,095	20.89	53.84	0.697	6	0
272	Undefined	Not Computed	66,195	80.65	44.55	−0.034	39	19

Legend: * Isoelectric point (pI).

**Table 5 viruses-15-00458-t005:** Molecular weight, theoretical isoelectric point (pI), extinction coefficient, aliphatic index, instability index, grand average of hydropathy (GRAVY), and total number of positive and negative residues of S protein.

	Molecular Weight	pI	Extinction Coefficient	Aliphatic Index	Instability Index	GRAVY	+ Residual AA	− Residual AA
800	20,315	8.20	70,315	97.99	53.87	0.650	6	3
799	17,234	8.38	63,075	107.87	51.98	0.894	6	3
772	19,625	8.46	67,210	91.79	58.99	0.653	7	3
738	19,232	8.46	68,700	96.86	64.12	0.708	7	3
489	20,018	8.45	70,190	92.49	63.90	0.596	7	3
454	20,267	8.58	67,335	91.45	60.25	0.619	8	3
384	20,248	8.54	67,335	91.45	63.84	0.615	8	3
320	20,305	8.57	68,825	95.81	58.70	0.634	8	3
170	18,885	8.46	70,190	97.47	54.11	0.798	7	3
069	20,089	8.76	67,210	90.28	53.71	0.557	9	3
001	20,292	8.46	68,700	93.63	59.40	0.641	7	3
272	Undefined	Not Computed	70,065	83.67	52.36	0.577	6	3

## Data Availability

Data are available on request to the Corresponding Author.

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
