# Peer review of "Molecular and Genetic Characterization of Hepatitis B Virus (HBV) among Saudi Chronically HBV-Infected Individuals"

_viruses, 2023, doi:10.3390/v15020458_

Round 1

Reviewer 1 Report

The study presented by Stefano et al is a study that performed molecular and genetic characterization (mutations) of HBV viral isolates in chronically infected individuals. While the study is interesting, it does not bring anything new to what is known about HBV genetial diversity for the region in question. Some major points need to be elucidated for a better understanding of the study population and methods. Here are some recommendations that may be pertinent to the study:

Abstract:

1 - "subtypes": the most used term and consensus is subgenotypes, it needs to be replaced throughout the text

2 - "Saudi patients were infected mainly by D genotype, mainly subtype 1": 

The authors can be more direct and say only subgenotype D1

3 - "The physicochemical properties of HBs proteins showed the presence of basic proteins":

Sentence a little confusing, no relevant information about the findings

Introduction

4 - This section could be improved and restructured, it is a bit confusing and needs more depth, to present the real problematic of the study

5 - "approximately 240 million people are chronically infected":

This information needs to be updated, according to data from the World Health Organization, this number is 290 million people (counting in 2019) (https://www.who.int/news-room/fact-sheets/detail/hepatitis-b).

Patients, Material and Methods

6 - The methodology in general needs to be better described

7 - It would be clearer and more fluid for the reader's understanding if this section were better divided into topics, for example, "study population; data collection; PCR; sequencing; end etc."

8 - "HBV serological markers (HBsAg, HBeAg, anti-HBs, anti-HBc, and anti-HBe) (using the Abbott Architect assay (Abbott diagnostics Division Max-Planck-ring 2, 65205, Wiesbaden, Germany). Serum HBV DNA viral load was determined by PCR-based assays (COBAS Amplicor, Roche Diagnostics, USA)."

It is unclear whether these serological and molecular tests are prior data collected from the patients' records or whether they are performed by the study, or whether they were done after the patients were selected

9 - "Plasma samples were obtained from 26 Saudi Arabian HBV-infected patients."

How were these samples obtained, after the recruitment of the patients? Were they fresh samples that were used and collected immediately after recruitment into the study? Or were they samples that were already stored previously in biobanks?

10 - "The real-time HBV PCR viral load was performed using the Artus® HBV RG PCR kit (QIAGEN GmbH, QIAGEN Strasse 1, 40724 Hilden, GERMANY)."

Why were the qPCR assays repeated, since the authors report that viral load quantification had been done previously? What is the time difference between the two tests performed, the first and the second?

11 - In what years were these patients recruited? what year are these samples from? this is not clear throughout the text

12 - "in-house PCR method as shown elsewhere":

Briefly describe the procedure of the in-house PCR technique. What is the detection limit of the technique?

13 - "For each patient, HBV-reverse transcriptase (HBV-RT) (344 amino acids) and the overlapped HBsAg (226 amino acids) full-length sequencing was performed on plasma samples as described previously":

It is not clear whether amplification and sequencing is performed for each fragment or whole genome. Why did the authors choose to perform this procedure on separate fragments, since the regions in question are overlapping in the viral genome?

14 - The sequencing performed in the study was Sanger sequencing? Make this fact clearer

Results 

16 - "Table 1"

The HBV-DNA results shown are from which qPCR test? the one done by COBAS Amplicor, Roche Diagnostics or QIAGEN?

17 - Why were only 12 samples selected for mutation and phylogenetic analysis? What was the criterion? From what was presented there were 26 plasma samples, and from the table 23 had detectable viral loads.

18 - It would be important to provide the phylogenetic trees performed, where they show the clustering of the study sequences with reference sequences

19 - How many refrerence sequences were used to assemble the phylogenetic tree? How many bootstraps were used to produce them? What are the criteria for defining the models of the substitutions?

20 - Isn't the fragment size used (gene polymerase) to establish HBV subgenotypes too small? According to several consensus statements in the literature, the optimal fragment size for establishing subgenotypes is at least 600 bp. 

21 - Have the sequences obtained in the study been deposited in genbank? Provide the access numbers

22  - As for the ethical issues of the study, has it been approved by an ethics committee? Please provide this information that is not clear in the text

Author Response

The study presented by Stefano et al is a study that performed molecular and genetic characterization (mutations) of HBV viral isolates in chronically infected individuals. While the study is interesting, it does not bring anything new to what is known about HBV genetial diversity for the region in question. Some major points need to be elucidated for a better understanding of the study population and methods. Here are some recommendations that may be pertinent to the study:

Abstract:

1 - "subtypes": the most used term and consensus is subgenotypes, it needs to be replaced throughout the text

R: The term was changed as requested

2 - "Saudi patients were infected mainly by D genotype, mainly subtype 1": 

The authors can be more direct and say only subgenotype D1

R: This was fixed 

3 - "The physicochemical properties of HBs proteins showed the presence of basic proteins":

Sentence a little confusing, no relevant information about the findings

R The suggestion was accepted and the sentence was removed

Introduction

4 - This section could be improved and restructured, it is a bit confusing and needs more depth, to present the real problematic of the study

R: We made all possible efforts to better structure the section. We hope that the referee will accept changes made

5 - "approximately 240 million people are chronically infected":

This information needs to be updated, according to data from the World Health Organization, this number is 290 million people (counting in 2019)

(https://www.who.int/news-room/fact-sheets/detail/hepatitis-b).

The suggestion was accepted and the text fixed accordingly

Patients, Material and Methods

6 - The methodology in general needs to be better described

R: We better described the methodology, as requested

7 - It would be clearer and more fluid for the reader's understanding if this section were better divided into topics, for example, "study population; data collection; PCR; sequencing; end etc."

R: The suggestion was accepted and the section fixed accordingly

8 - "HBV serological markers (HBsAg, HBeAg, anti-HBs, anti-HBc, and anti-HBe) (using the Abbott Architect assay (Abbott diagnostics Division Max-Planck-ring 2, 65205, Wiesbaden, Germany). Serum HBV DNA viral load was determined by PCR-based assays (COBAS Amplicor, Roche Diagnostics, USA)."

It is unclear whether these serological and molecular tests are prior data collected from the patients' records or whether they are performed by the study, or whether they were done after the patients were selected

R: Patients were known as HBV infected and serological data were obtained from records but they were repeated when they were evaluated for possible antiviral treatments. Molecular data were then obtained.

9 - "Plasma samples were obtained from 26 Saudi Arabian HBV-infected patients."

How were these samples obtained, after the recruitment of the patients? Were they fresh samples that were used and collected immediately after recruitment into the study? Or were they samples that were already stored previously in biobanks?

R:The plasma samples were collected at the time of evaluation of the patients for possible antiviral treatment. Accordingly to the type of test, samples were either used fresh or stored frozen until use.

10 - "The real-time HBV PCR viral load was performed using the Artus® HBV RG PCR kit (QIAGEN GmbH, QIAGEN Strasse 1, 40724 Hilden, GERMANY)."

Why were the qPCR assays repeated, since the authors report that viral load quantification had been done previously? What is the time difference between the two tests performed, the first and the second?

R: Right: this information was due to a mix-up in the text. We now fixed it.

11 - In what years were these patients recruited? what year are these samples from? this is not clear throughout the text

12 - "in-house PCR method as shown elsewhere":

Briefly describe the procedure of the in-house PCR technique. What is the detection limit of the technique?

R: The method is now described and the detection limit (20 copies/ml) is now reported.

13 - "For each patient, HBV-reverse transcriptase (HBV-RT) (344 amino acids) and the overlapped HBsAg (226 amino acids) full-length sequencing was performed on plasma samples as described previously":

It is not clear whether amplification and sequencing is performed for each fragment or whole genome. Why did the authors choose to perform this procedure on separate fragments, since the regions in question are overlapping

R: We adopted the methodology by Svicher et al (2009). The sequencing was performed on one fragment, that included both polymerase and HBs. 

14 - The sequencing performed in the study was Sanger sequencing? Make this fact clearer

R: Yes, it was Sanger method. We added now this in the text

Results 

16 - "Table 1"

The HBV-DNA results shown are from which qPCR test? the one done by COBAS Amplicor, Roche Diagnostics or QIAGEN?

R: Right, data were confusing. The reults shown in table 1 were obtained by Quiagen and this is now clarified in the text

17 - Why were only 12 samples selected for mutation and phylogenetic analysis? What was the criterion? From what was presented there were 26 plasma samples, and from the table 23 had detectable viral loads.

R: There were not selection criteria for this: some technical problem probably did not allow us to perform the analysis on some samples

18 - It would be important to provide the phylogenetic trees performed, where they show the clustering of the study sequences with reference sequences

R: We added now the philogenetic trees. The referee can choose if include them in the paper or to use them as supplementary figures

19 - How many refrerence sequences were used to assemble the phylogenetic tree? How many bootstraps were used to produce them? What are the criteria for defining the models of the substitutions?

R: As stated in the manuscript reference sequences were used according to "genotype reference sequences as recommended by Schaefer (ref 19 in the manuscript) as well as the closest previously published sequences in GenBank identified by BLASTN". Schaefer used one reference sequence per subgenotype, labelled in the trees as "GT-X.accession number", plus a non-human primate virus as outgroup; 7+1 sequences. In addition, BLASTN identified the closest previously published sequences in GeneBank, that were added to the phylogeny;sometimes the same GeneBank sequence was identified as closest to >1 of our sequences, thus this resulted in adding 11 and 7 GeneBank sequences to the S and RT trees. We did not use Bootstrap, but rather the SH-test in PhyML to assess reconstruction robustness. Notably all GeneBank sequences were subgenotyped as genotype D1 in agreement with their original classification, providing adequate subgenotyping robustness. While the nucleotide model has little impact on genotype classification, we used GTR+I+G as it best describes HBV evolution in general

20 - Isn't the fragment size used (gene polymerase) to establish HBV subgenotypes too small? According to several consensus statements in the literature, the optimal fragment size for establishing subgenotypes is at least 600 bp. 

R: S fragment alignment was 536 nt and RT was 831 nt. These lenghts were adequate for subgenotyping, as the trees from these alignments were robust and confirmed both the Schaefer and GenBank reference sequences subgenotype classifications.

21 - Have the sequences obtained in the study been deposited in genbank? Provide the access numbers

R: We do agree with this observation. Surely it will be done in the due time.

22  - As for the ethical issues of the study, has it been approved by an ethics committee? Please provide this information that is not clear in the text

R:The patients signed a written informed consentin agreement with WMA Helsinki declaration allowing the use of  blood smples for analysis aimed to monitor the planned antiviral treatment , including biomolecular studies.

All the lab test included in the paper, in fact, are part of the bioclinical assessment and follow up of antiviral treatments given, or planned to be given, to the patiants.

Due to the purposes, then, ethical committee approval was not necessary

Reviewer 2 Report

Were the patients all of the same ethnicity and region of origin (South Arabia is a very large country and is close to other populous countries in the Middle East, Africa and Asia)?

In other words, my problem is whether the relatively small sample, yet so deeply and excellently examined by you, is well representative of the entire population of the area/country. In any case, can be useful specify about.

I ask the Authors to add, if possible, a few more considerations that correlate the results obtained and the problem of non-responders to the HBV vaccine. The authors mention this crucial theme only one on page 6: "escape from vaccine-induced immunity." If they follow my suggestion, please also mention the issue in the discussion and abstract.

Transient elastography (TE)

TE in parenthesis immidiatly after Transient elastography.

Author Response

Were the patients all of the same ethnicity and region of origin (South Arabia is a very large country and is close to other populous countries in the Middle East, Africa and Asia)?

In other words, my problem is whether the relatively small sample, yet so deeply and excellently examined by you, is well representative of the entire population of the area/country. In any case, can be useful specify about.

R Yes, all patients were Saudi nationals and this is now stated

I ask the Authors to add, if possible, a few more considerations that correlate the results obtained and the problem of non-responders to the HBV vaccine. The authors mention this crucial theme only one on page 6: "escape from vaccine-induced immunity." If they follow my suggestion, please also mention the issue in the discussion and abstract.

R We did not focus that much on non responders to HBV vaccines because

  1. None of our patients had been vaccinated
  2. We are planning further studies on subjects non responder to HBV vaccinae

We did not want to spend words on pure speculations and we hope that, in spite of the importance of the observation, still the referee will accept it

Transient elastography (TE)

TE in parenthesis immidiatly after Transient elastography.

R this was fixed

Round 2

Reviewer 1 Report

Many of the questions have been answered by the authors, but many important points still need to be answered and elucidated.

1 - In which years were these patients seen?

2 - The authors should make it clearer throughout the text that the data collected from the medical records have been redone, for example the serology results, as questioned earlier.

3 - It would be important for the authors to provide in the methodology section the same explanation given to question 19, previously requested.

4 - The phylogenetic trees presented are not very clear, not showing clearly the results of the study in comparison to the reference samples. In addition, the authors should provide a self-explanatory legend for both figures to provide readers with clearer information about the data presented.

5 - In table 1, the authors still need to make clear in the text, which test was used for the HBV DNA data presented, showing that the test was the Quiagen test, as answered earlier, this information is still not in the text as reported.

6 - The fact that only 12 samples were used for the analysis still needs to be presented more clearly throughout the text. The authors report technical problems, as previously answered, but what were these technical problems? As a suggestion, this information could be added in a paragraph about the limitations of the study.

Author Response

1 - In which years were these patients seen?

R. This is now stated in the text

2 - The authors should make it clearer throughout the text that the data collected from the medical records have been redone, for example the serology results, as questioned earlier.

R. we clarify this: there was a mistakebecause tests were not redone.

3 - It would be important for the authors to provide in the methodology section the same explanation given to question 19, previously requested. 

R. We did it according to referee request.

4 - The phylogenetic trees presented are not very clear, not showing clearly the results of the study in comparison to the reference samples. In addition, the authors should provide a self-explanatory legend for both figures to provide readers with clearer information about the data presented.

R. We fullfilled the request of the referee.

5 - In table 1, the authors still need to make clear in the text, which test was used for the HBV DNA data presented, showing that the test was the Quiagen test, as answered earlier, this information is still not in the text as reported.

6 - The fact that only 12 samples were used for the analysis still needs to be presented more clearly throughout the text. The authors report technical problems, as previously answered, but what were these technical problems? As a suggestion, this information could be added in a paragraph about the limitations of the study.

R. We agree with the referee: the issue is now indicated as a limit of the study.